# Horizontal and Uplift Bearing Characteristics of a Cast-In-Place Micropile Group Foundation in a Plateau Mountainous Area

**Haitao Li and Guangming Ren ***

State Key Laboratory of Geohazard Prevention and Geo-Environment Protection,
Chengdu University of Technology, Chengdu 610059, China; 2021050213@stu.cdut.edu.cn
* Correspondence: rengmgcr@163.com; Tel.: +86-188-4263-8422

**Abstract:** Micropile groups have been progressively more frequently adopted in the construction of transmission tower bases due to their compact size and flexible construction advantages. However, the load-bearing characteristics and deformation mechanisms of micropile groups are complex, and the study of their coupling effects under combined loads remains unclear. Consequently, this paper presents a field static load test of micropile groups in a highland mountainous area. The analysis encompasses the axial force distribution and load-sharing ratio of micropiles. With a focus on micropile groups subjected to both uplift and horizontal combined loads, the coupled effects under different load combination ratios are examined using numerical simulation methods. The key findings are as follows: During the uplift loading process, the load distribution among individual piles is relatively uniform, with lower side friction resistance gradually coming into play to counterbalance the top load. The load–uplift displacement curve exhibits a steep characteristic, making it susceptible to sudden failure in practical engineering applications. Under the simultaneous action of uplift (V) and horizontal (H) loads, the unbalanced lateral frictional resistance on both sides of the pile segment induces additional bending moments, which is an important part affecting the load-coupling mechanisms. The uplift resistance capacity of micropile groups decreases with an increase in horizontal load, while the horizontal load-carrying capacity initially decreases and then increases with an increase in uplift load. The space enclosed by the yield envelope under combined load, and the vertical line of the ultimate load, is divided into a 'failure zone' and a 'safety zone.' In the design of the pile foundation, the uplift bearing capacity reduced by the 'failure zone' should be taken into account.

**Keywords:** micropile group; plateau mountainous area; friction resistance of pile; load-coupling effect; pile–soil interaction

## 1. Introduction

With the rapid expansion of power projects in Western China, there is a growing demand for transmission and transformation projects in mountainous terrain. Traditional transmission lines typically rely on large-volume cast-in-place pile foundations, which can be expensive and pose challenges in terms of quality control. Furthermore, in regions with complex environmental conditions, transporting the necessary equipment for cast-in-place pile construction can be logistically challenging. Therefore, while ensuring the stability of foundations remains crucial, there is a pressing need to minimize material usage and streamline construction processes. Micropile technology, known for its compact size and flexible construction methods, has traditionally been employed in foundation reinforcement and slope stabilization [1–3]. However, as foundation construction techniques have advanced, micropiles have gradually found applications in power transmission line projects [4,5] and have even been used in selected soft soil and loess foundations. These developments position micropiles as one of the innovative technologies endorsed by the State Grid Corporation of China, offering promising prospects for wider adoption and implementation.



In recent years, researchers both domestically and internationally have conducted extensive investigations into the force mechanisms of micropiles, employing prototype tests, model tests, and numerical simulations, and yielding noteworthy findings. Theoretical analyses of micropiles typically rely on three primary methods: the p–y curve method [6–8], the load–structure method [9], and the finite element method [10–12]. All these approaches serve for both qualitative and quantitative assessments of micropile bearing capacity. However, while the former two methods involve complex derivations, the finite element method stands out for its suitability in simulating the intricate interactions between micropiles and the surrounding soil under diverse loading conditions. This approach enables a comprehensive analysis, encompassing multiple influencing factors. As micropile technology continues to advance, several studies have underscored the significant impact of various parameters, including soil density and installation methods, on the load-bearing characteristics of micropiles [13,14]. The soundness and feasibility of micropile construction techniques are paramount for the successful execution of engineering projects. Researchers have delved into the suitability and efficacy of various construction methods and technologies across diverse geological conditions. They have conducted comprehensive analyses of the mechanical behavior of micropile groups subjected to both vertical and lateral loading scenarios, with the aim of offering robust design and construction recommendations [15,16]. Furthermore, in terms of micropile performance, Zhang et al. [17] carried out in situ static load tests to evaluate the resistance to uplift of micropiles in soft red clay substrates. Their findings indicate that micropiles can notably enhance pull-out resistance and effectively control foundation uplift deformation. Meanwhile, Murthy et al. [18], utilizing numerical simulations, explored the settlement characteristics of foundations reinforced with micropiles, ultimately concluding that micropile reinforcement significantly reduces foundation settlement.

Nevertheless, with the widening spectrum of micropile applications, conventional construction methods for micropiles face difficulties in addressing intricate construction needs. A conspicuous challenge arising from their diminutive diameter is the diminished lateral load-bearing capacity. Consequently, to meet the load-bearing prerequisites, it is customary to utilize multiple micropiles, which are interconnected into an integrated system using sturdy beams or plates at the pile heads [19]. Consequently, a complex foundation structure is formed, comprising micropile foundations, upper support platforms, and the surrounding geological formations. In the study of micropile groups, it is customary to take into account factors such as pile spacing, arrangement patterns, and the influence of pile group effects to ensure the stability and uniformity of micropile foundations [20]. Researchers have investigated the load-bearing performance of micropile groups under both vertical and horizontal loads, as well as the group effects, through a combination of field and model tests [21–24]. However, the test results have exhibited contradictions, showing variations in the load-bearing performance of micropile groups. The primary reason for these discrepancies lies in the fact that the load behavior of micropile groups in the experiments occurred within different characteristic soil conditions. Therefore, further refinement of the study of micropile load behavior is required, particularly in complex soil environments.

The adaptability of micropile groups to different sites is particularly crucial. Concerning seismic behavior, micropile groups exhibit flexible behavior in terms of soil–foundation–structure interactions [25]. During dynamic response processes, they dissipate a significant amount of energy, effectively preventing brittle failure and demonstrating good seismic performance. Studies have indicated that micropile groups provide effective support for seismic dynamic response in liquefiable soil conditions [26]. Simultaneously, micropile groups offer notable advantages in addressing sloping ground situations [27]. Their flexible reinforcement characteristics significantly enhance the soil's shear resistance, mobilizing the soil's resistance to pullout and sliding, thereby forming a combined support system that enhances the overall stability of a sloping terrain [28].

Borthakur and Dey [29] conducted a systematic investigation into micropile groups in soft soils. Their study involved an analysis of the nonlinear relationship between the load-bearing capacity of micropile groups and various parameters, including pile diameter, length, quantity, and spacing. Additionally, they conducted a comparative analysis of the effects of caps positioned close to the ground surface. Their research findings highlighted a significant enhancement in the compressive load-bearing capacity of micropile groups when caps were situated near the ground surface. Furthermore, they emphasized that pile spacing had the most pronounced impact on the load-bearing capacity of micropile groups, corroborating similar results from Zeng et al.'s study [30]. Zeng and colleagues also noted that with equivalent pile circumferences, the construction process resulted in a higher ultimate load-bearing capacity for drilled and grouted micropile groups compared to precast micropile groups, along with distinctive settlement characteristics. Similarly, Du et al. [31] conducted model experiments in conjunction with numerical simulations to analyze the load-bearing characteristics of intensive micropile groups. They also investigated the sensitivity of the horizontal load-carrying performance of pile groups to variations in design parameters such as soil properties between piles, pile strength, and pile spacing. Furthermore, Zhang [32] studied the load characteristics of low-cap micropile groups based on the differential equations for the deflection of an elastic foundation beam. Zhang proposed an improved calculation method for low-cap micropile groups [32]. Regarding the failure modes of micropile groups, the study conducted by Xu et al. [33] indicates that under vertical uplift and horizontal loading, the group pile foundations exhibit non-integral failure modes and conical wedge failure modes, respectively. On the other hand, research by Hussain et al. [34] suggests that the failure mode of micropile groups is a function of the soil relative density and the aspect ratio. Micropile groups with a larger aspect ratio tend to have higher load-carrying capacity.

The aforementioned studies have provided valuable contributions to research on micropile load-bearing characteristics and their engineering applications. However, at the current stage, there is a limited number of field validation tests for micropile groups in high-altitude mountainous regions. Additionally, micropile research often focuses on individual loads, leaving the understanding of the load-bearing characteristics of micropile groups under the influence of complex combined loads unclear. Therefore, this study conducts prototype experiments on micropile groups in high-altitude mountainous regions to validate their load-bearing performance. It analyzes the distribution characteristics of axial forces and lateral frictional resistance, and the load-sharing ratio, among various components. Furthermore, considering the scenario where micropile groups simultaneously bear uplift and horizontal combined loads, and building upon the validation of the accuracy and reliability of numerical models, this research simulates the condition where the ratio of the vertical and horizontal loads (U–H) is 2:1. It investigates the combined action mechanisms of micropile groups through the analysis of pile body strain and lateral frictional resistance. Additionally, it performs load-carrying capacity and load envelope analysis on pile foundations under different uplift-to-horizontal-load ratios, revealing the coupling effects of U–H combined loads. This research contributes valuable insights to the application and improvement of micropile groups.

## 2. Field Testing of Micropile Group Foundations

### 2.1. Experimental Overview

The experimental site is located in the gently sloping plateau region of Aba Prefecture, Sichuan Province, China, at an elevation exceeding 2000 m. The area features a widespread and moderately thick deposit of gravelly silty soil, with some tower foundations directly situated onto this soil layer. Based on the field exploration results (Figure 1), the geological strata within the scope of the site investigation exhibit relative simplicity, primarily comprising two layers. These layers are identified as the Quaternary silty gravel and silty sand deposits. Samples were collected from two types of foundation soils exposed during drilling, and laboratory geotechnical tests were conducted (Figure 2). The physical and

mechanical properties of various strata within the site were determined, as presented in Table 1. From Table 1, it can be observed that the basic physical properties of the foundation soils do not vary significantly with depth. However, there are spatial variations in the physical and mechanical properties. The particle size distribution curves obtained from the test results are shown in Figure 3. The upper layer consists of gravelly soil with a gravel content greater than 50%, and the fines content (particles smaller than 0.075 mm) is 15.56%. The fines content is higher than the clay content, with a Cu (coefficient of uniformity) equal to 79 and a Cc (coefficient of curvature) equal to 1.034, indicating well-graded gravelly silt. The lower layer mainly comprises sandy soil, with a fines content of 30.13%. Similar to the upper layer, the fines content exceeds the clay content, and the soil is poorly graded sandy silt, with Cu = 89 and Cc = 0.625.

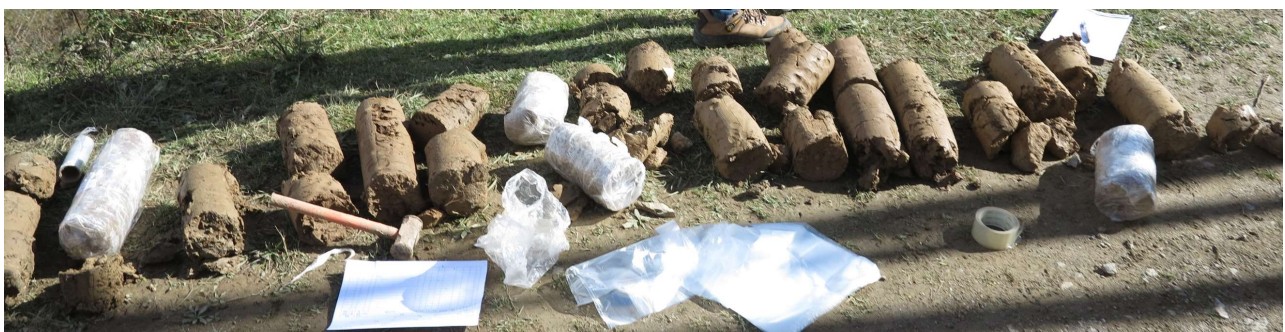

**Figure 1.** Partial core photographs revealed through on-site exploration.

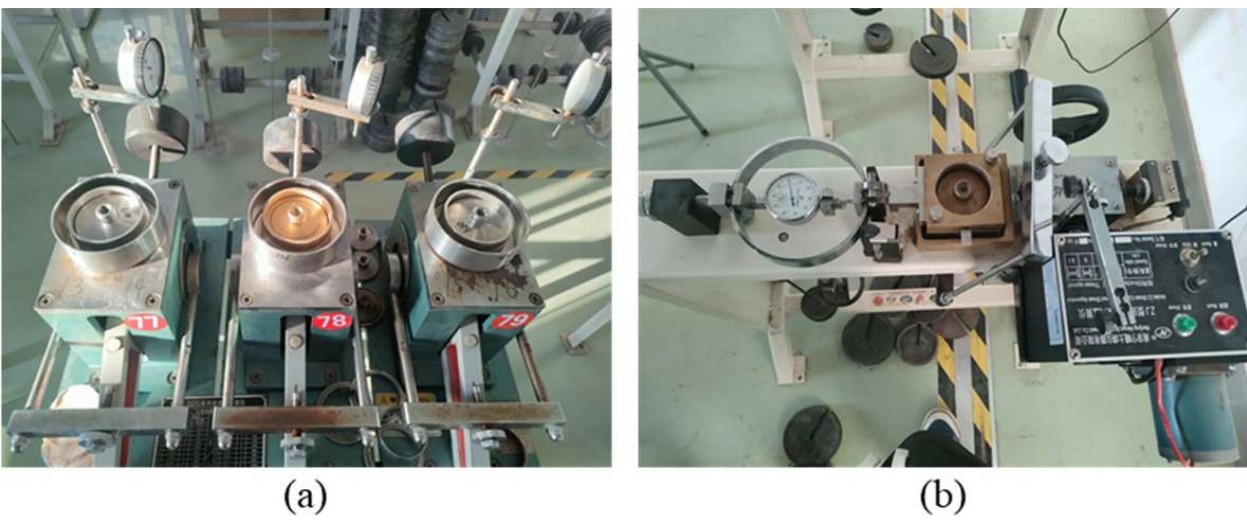

**Figure 2.** Selected laboratory geotechnical tests: (**a**) consolidation test; (**b**) direct shear test.

**Table 1.** Soil parameters of the test site.

| Sample ID | Sampling Depth/m | Water Content/% | Plasticity Index | Compression Modulus /MPa | Poisson Ratio | Cohesion /kPa | Internal Friction Angle/(°) |
|---|---|---|---|---|---|---|---|
| 1 | 0~1.6 | 22.94 | 12.5 | 14.9 | 0.33 | 12.50 | 25.60 |
| 2 | 1.6~3.0 | 20.12 | 13.0 | 15.3 | 0.32 | 14.89 | 28.45 |
| 3 | 3.0~3.9 | 19.89 | 11.6 | 18.8 | 0.30 | 20.47 | 15.05 |
| 4 | 5.2~5.8 | 22.18 | 14.5 | 19.4 | 0.29 | 23.08 | 17.39 |
| 5 | 5.8~6.2 | 21.86 | 14.8 | 21.5 | 0.30 | 23.61 | 17.87 |
| 6 | 8.0~9.0 | 24.43 | 15.1 | 15.4 | 0.32 | 17.18 | 30.49 |

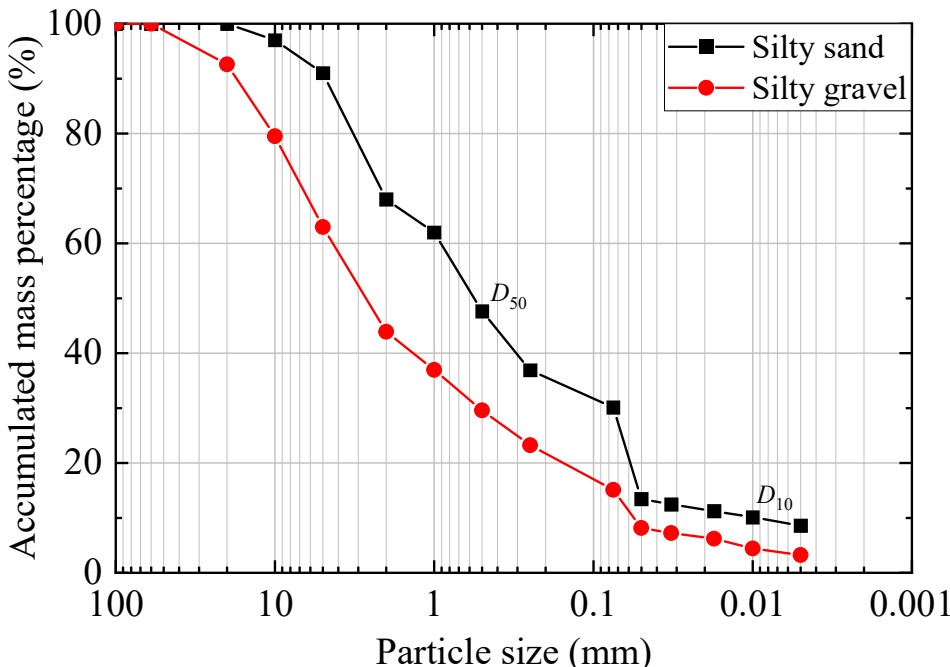

**Figure 3.** Grain-size analysis curve.

The test piles in this study are micropile group foundations with uniform sections, composed of three components: the pile shaft, cap, and upper columns. Each individual pile measures 7 m in length and 0.3 m in diameter, spaced at intervals of 3 d (where d represents the diameter). The pile cap has dimensions of 1.8 m by 1.8 m by 0.6 m, while the upper columns measure 1.0 m by 1.0 m by 1.5 m. The structural arrangement of the micropile group foundation is illustrated in Figure 4. C35 grade concrete was utilized for construction, reinforced with HRB400 hot-rolled deformed steel bars, 14 mm in diameter, for the main reinforcement, and HPB300 hot-rolled plain round steel bars, 8 mm in diameter, for the stirrups.

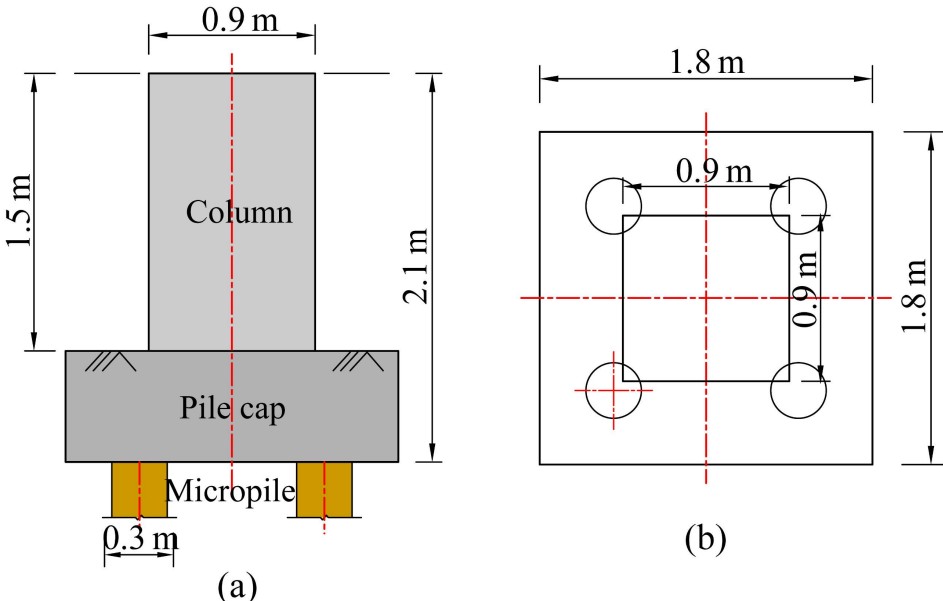

**Figure 4.** Micropile group foundation structure and reinforcement: (**a**) side view; (**b**) plan view.

The test piles were constructed using a small rotary drilling rig and cast through the method of auger drilling and grouting. After casting, the quality of the micropile

installation was inspected, and a cap was cast from the top of the pile to a depth of 0.6 m. The on-site construction process is illustrated in Figure 5. To ensure the monitoring of axial forces at different depths of the micropiles (1 m below the pile top, mid-section, and 1 m above the pile bottom), three strain gauges were installed beneath the pile cap of each pile. This setup allowed for the monitoring of axial forces at the top, middle, and bottom sections of the micropiles, ensuring the monitoring of axial force distribution and variations in different soil layers. The specific locations of the strain gauge installations are shown in Figure 6.

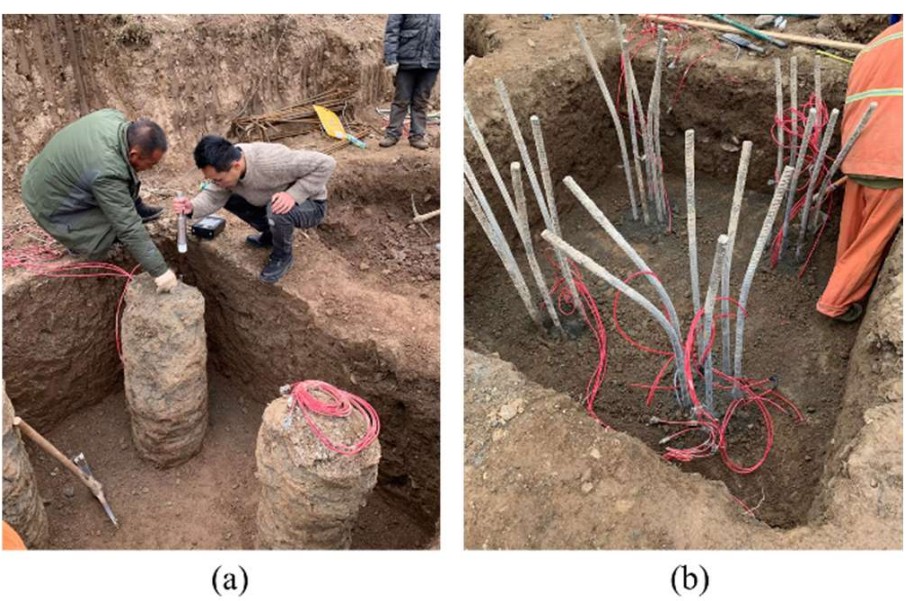

**Figure 5.** Field construction: (**a**) pile quality check; (**b**) pile cap pouring.

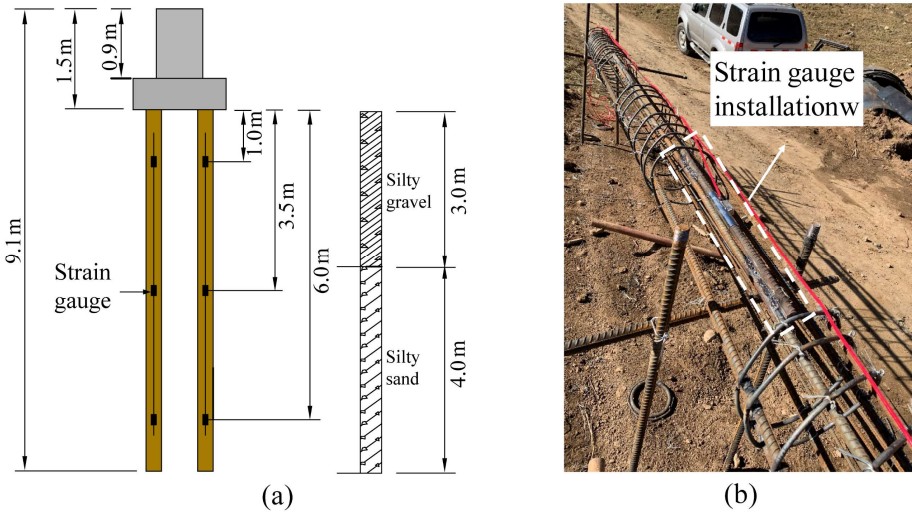

**Figure 6.** Stressometer layout: (**a**) schematic diagram; (**b**) site operation.

The test was conducted using reaction piles with a diameter of 1.6 m and a length of 8 m to provide support reaction, and the micropiles were constructed using the method of manual auger drilling and grouting. The experimental loading setup consisted of two 1000 kN capacity jacks placed on the reaction piles, and four LVDTs (linear variable differential transformers) with a measuring range of 50 mm were positioned on the pile cap surface. Vertical displacements at the pile cap were measured at each load increment, with the LVDTs placed at the edges of the upper column plane. The reaction piles were positioned on both sides of the test pile and transmitted force through steel beams, which were further

transferred to the test pile cap via tension rods. Two separate tests were conducted on-site: an uplift resistance test (GMP–1) and a horizontal resistance test (GMP–2). The experimental loading setup and on-site layout are illustrated in Figure 7.

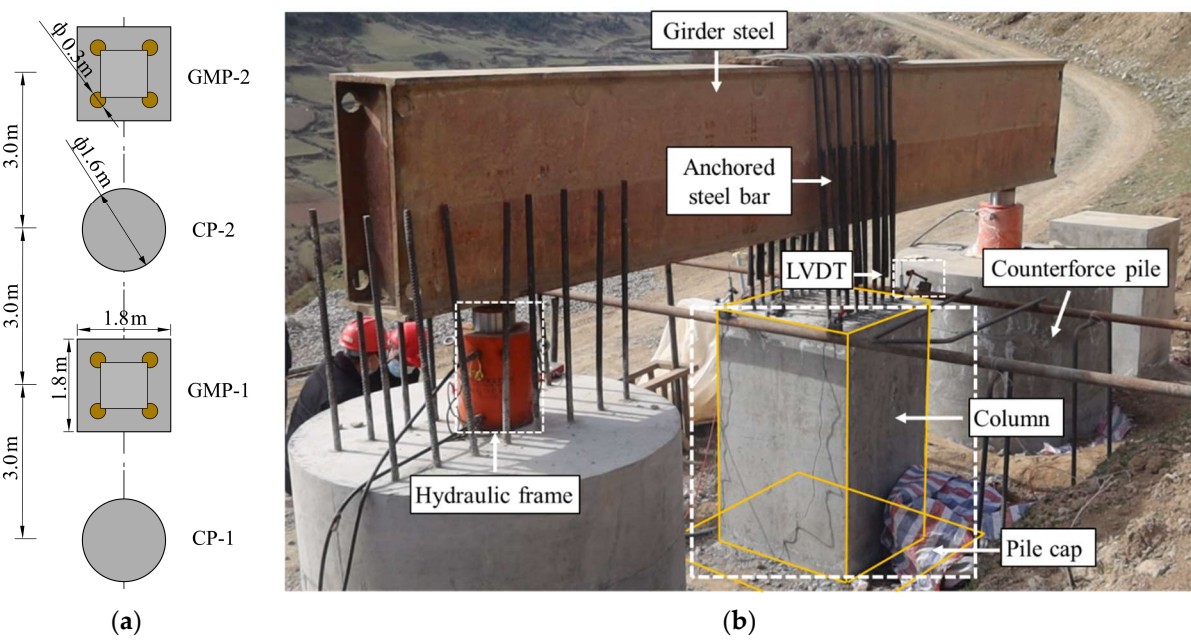

**Figure 7.** Loading test of the micropile group: (**a**) plan view of test site (**b**) pull–out test site.

### 2.2. Experimental Loading Process and Result Analysis

The experiment was conducted in accordance with the "Technical Code for Testing of Building Foundation Piles" (JGJ106–2014: China Architecture and Buliding Press, Beijing, China) using the slow sustained load method to apply loads on the group pile foundation. The estimated ultimate uplift bearing capacity of the group pile foundation was 2000 kN, and the estimated ultimate horizontal bearing capacity was 1000 kN. The load increments were set as 1/10 of the estimated ultimate loads, resulting in loading increments of 200 kN for uplift tests and 100 kN for horizontal static load tests. For the uplift loading, the first load increment was set at 2 times the standard load increment. During the experiment, the loading was stopped when the foundation reached failure. The failure criteria were defined as the point where the load dropped sharply or when the uplift displacement reached 30 mm. For the horizontal direction, the horizontal bearing capacity of the pile was determined at a displacement of 10 mm (or 6 mm for buildings sensitive to horizontal displacement) at the ground level. The test results (Figure 8) showed that for GMP–1, the load–displacement curve exhibited a sudden change at the 6th load increment. This observation suggests the occurrence of relative sliding between the pile and the soil at this point, with a critical uplift capacity limited to only 1200 kN. As for GMP–2, the load–horizontal displacement curve was relatively gentle, and the horizontal ultimate bearing capacity of the micropile group foundation was determined to be 620 kN, corresponding to a horizontal displacement of 10 mm. The on-site test results were compared with the results obtained using the simplified calculation method recommended in the "Technical Code for Building Pile Foundations" (JGJ94-2008: China Industy Building Press, Beijing, China). The comparative results are presented in Table 2. It is evident from Table 2 that the two methods yield similar bearing capacity values. However, the results obtained using the traditional calculation method are slightly lower than the on-site test results. This difference may be attributed to the increased group pile efficiency factor in the micropile group [35].

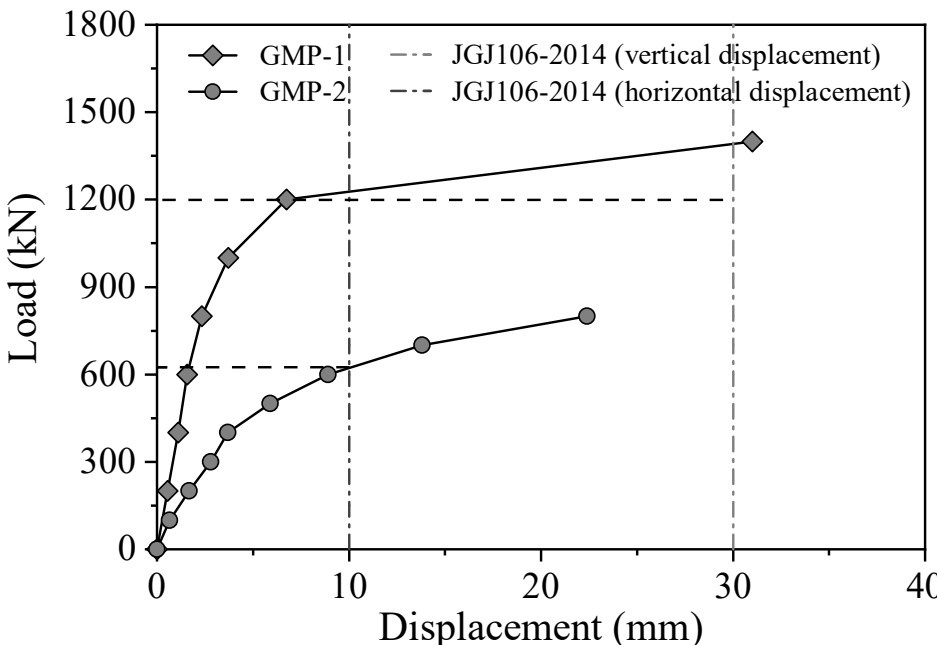

**Figure 8.** Load displacement curve.

**Table 2.** Comparison between test results and standard calculation methods.

| Type of Load | Field Test (kN) | Simplified Standard Calculation Method (kN) | Deviation Rate |
|---|---|---|---|
| Uplift Load | 1200 | 1109 | 7.6% |
| Lateral Load | 620 | 596 | 3.9% |

Note: Deviation rate calculated as a percentage difference between field test results and simplified standard calculation method.

As the main purpose of this horizontal test was to verify the horizontal bearing capacity of the micropile group foundation, displacement measurements were only taken on GMP–2 at different horizontal loads without applying strain gauges to its pile shaft. Therefore, this study does not include analysis of GMP–2's pile shaft displacements and bending moments. The load at the pile cap is assumed to be uniformly distributed into four parts, each acting on the upper section of the corresponding micropile. The axial force and lateral frictional resistance distribution along the depth of the pile shaft for each component are illustrated in Figures 9 and 10, respectively.

According to the locations of the stress gauges, the depth of the micropile foundation was divided into three parts: Part 1 (from the ground level to a depth of 1.6 m), Part 2 (from a depth of 1.6 m to 4.1 m), and Part 3 (from a depth of 4.1 m to 6.6 m). From Figure 9, it can be observed that the axial force curves of the four piles show slight variations in numerical values, but overall follow a consistent pattern: when the uplift load is relatively small, the axial force distribution along the pile shaft is relatively uniform. As the top load increases, the axial force gradually diminishes along the depth direction. Part 3 exhibits a larger variation in axial force, and this variation increases with the increment of load. Part 1 also shows a slightly increased variation in axial force, while Part 2 experiences a smaller change. The variation in the slope of the axial force curve corresponds to the development of lateral friction resistance. This observation aligns with the findings of Xu et al.'s indoor model tests [33], which indicate that, under the influence of early-stage uplift loads, the upper section of the pile shaft often mobilizes its resistance earlier than the lower section.

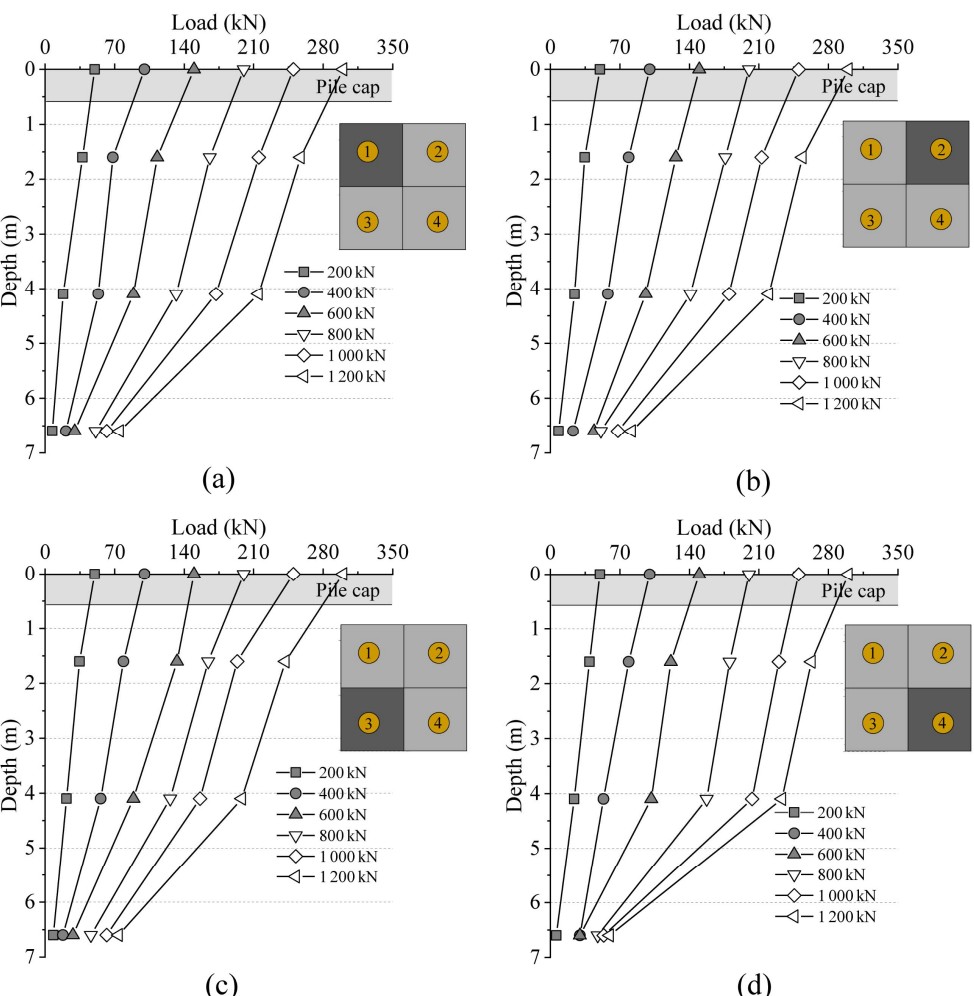

**Figure 9.** Pile axial force distribution curve: (**a**) foundation pile 1; (**b**) foundation pile 2; (**c**) foundation pile 3; (**d**) foundation pile 4.

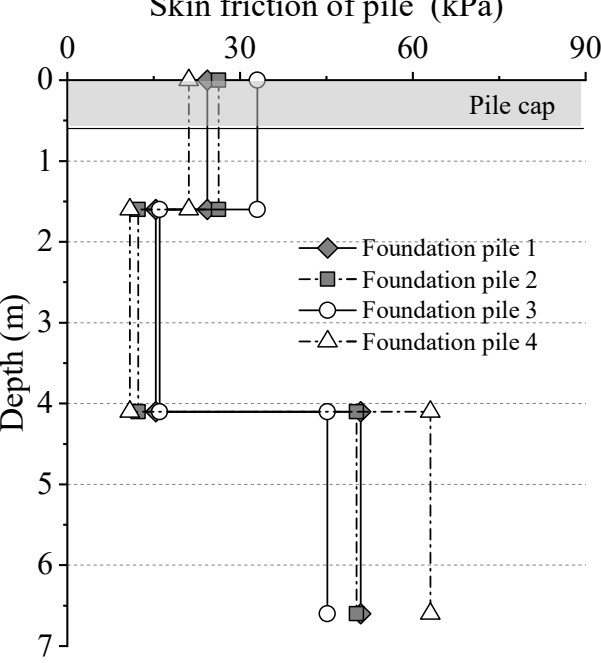

**Figure 10.** Pile side friction resistance curve.

As the load increases, the lateral resistance on the upper part of the pile decreases, and the lower lateral friction resistance gradually comes into play to counterbalance the top load until reaching the ultimate state. This indicates that the transfer of lateral friction resistance increases gradually from top to bottom. As shown in Figure 10, at the ultimate load, Part 3 exhibits the highest average lateral friction resistance (45–63 kPa), followed by Part 1 with a slightly lower average lateral friction resistance (21–33 kPa), and Part 2 shows the lowest average lateral friction resistance (11–16 kPa).

Considering the load distribution among different sections of the piles and plotting the load distribution curve, as shown in Figure 11, it becomes evident that when a load of 200 kN is applied at the pile head, the three sections of the piles generally share the load equally. As the load at the pile head increases, the load distribution among these sections also increases. Notably, Part 3, situated at the lower part of the piles, experiences a more significant increase in load distribution. This is attributed to the fact that the top load gradually transfers to this section. Furthermore, due to the higher compressive modulus (greater stiffness) of the lower soil, the ratio of compressive modulus Eb to Es between the lower soil and the soil around the pile (Eb/Es) is relatively small. This results in the accumulation of lateral frictional resistance along this portion of the pile [36]. Consequently, this section experiences the most significant increase in load distribution, reaching 530 kN at the ultimate load. Its load distribution accounts for 61% of the ultimate uplift capacity of the entire micropile group. In contrast, Part 1 and Part 2 share a relatively similar load distribution, with load distributions of 150 kN and 185 kN, respectively, at ultimate load. Their load distributions represent 18% and 21%, respectively, of the total capacity. This indicates that when the micropile group is subjected to uplift loads alone, the contribution of the pile cap is not significant, and it does not substantially enhance the uplift capacity of the foundation.

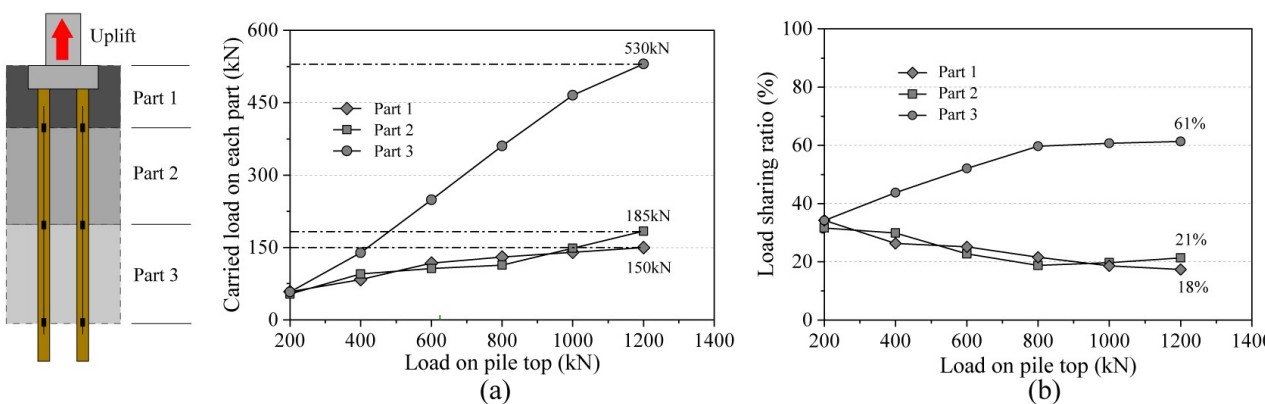

**Figure 11.** Load-sharing behavior of each part of micropile group foundation: (**a**) carried load by each part of micropile group foundation under loading; (**b**) load-sharing ratio of each part of micropile group foundation under loading.

## 3. Numerical Simulation Study of the Micropile Foundation

To verify the reliability of the experimental results and comprehensively analyze the bearing characteristics of the micropile group foundation in the plateau mountainous area, as well as to achieve visualization of deformations, this section conducts numerical simulations based on the agreement with the on-site test results. The study focuses on investigating the combined influence of the uplift and horizontal loads at different proportions on the bearing characteristics of the micropile group foundation.

### 3.1. Establishment of Numerical Model

Taking into account the influence of pile length and the effect of the pile cap, the geotechnical soil extends 6.6 times the length of the pile cap in both the x and y directions, and 2 times the length of the pile in the z direction. The model dimensions are 12 × 12 × 15 m. The constitutive model for the soil adopts the Mohr–Coulomb model,

which includes non-associated flow criteria. The pile's constitutive behavior is modeled using an isotropic elastic–plastic model. Both the soil and pile are simulated using solid finite elements, and their interaction is represented by establishing interface elements between the soil and the pile foundation [37]. Normal displacement constraints are applied on the model's lateral sides, fixed constraints are applied at the bottom, and the model's top surface is considered a free surface. Due to the low groundwater level at the test site, the influence of groundwater is not considered during the simulation. To balance computational efficiency and accuracy, the mesh is refined more extensively in the pile–soil contact region and becomes relatively coarser when further away from this area, with an increment space geometric ratio set to 1.2.

The cohesive strength (c) and friction angle ($\varphi$) of the interface elements can be chosen as 0.5 to 0.8 times the corresponding values of the soil around the pile (0.8 for cast-in-place piles and 0.5 for precast piles). In this study, a value of 0.8 is selected. The shear stiffness ($k_s$) and normal stiffness ($k_n$) for the interface elements are determined using Equation (1). The geotechnical soil is divided into two layers: a gravelly silty layer from 0 to 3 m deep and a silty sandy layer below 3 m. The soil model parameters are selected as the average values of the respective layer's soil parameters. To address convergence issues related to maximum unbalanced forces, the soil layers and pile foundation model are generalized as shown in Figure 12. The numerical simulation parameters for the soil, micropile group, and interface are presented in Tables 3 and 4.

$$k_n = k_s = 10\max\left(\frac{K + 4G/3}{\Delta Z_{\min}}\right) \tag{1}$$

where $K$ and $G$ represent the bulk modulus and shear modulus of the elements near the contact interface, respectively, and $\Delta Z_{\min}$ denotes the minimum size of the normal connection region of the contact interface.

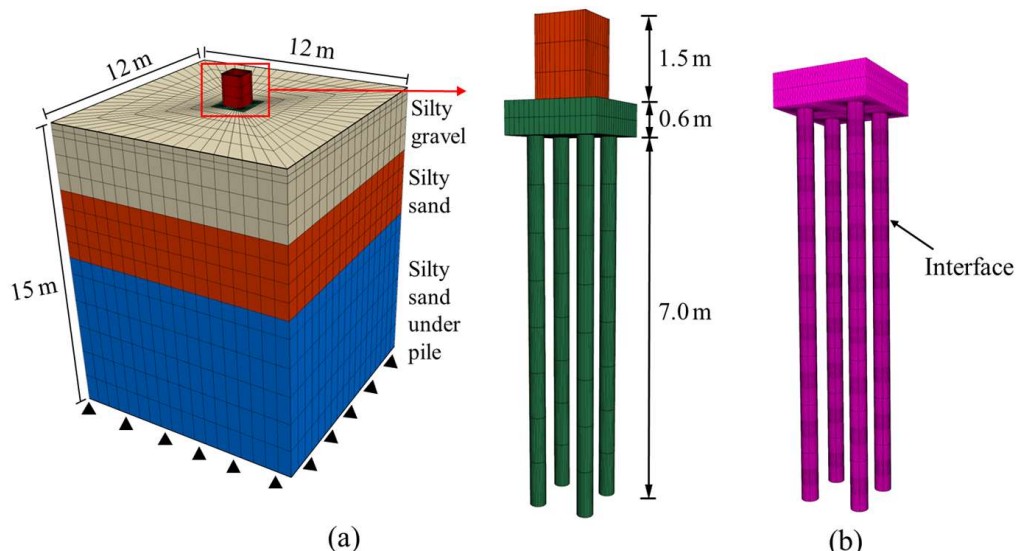

**Figure 12.** Numerical model of micropile group foundation: (**a**) the model of foundation soil and micropile groups; (**b**) interface model.

During the model calculations, two analysis steps were performed. The first step involved stress equilibrium in the ground. It was necessary to establish the initial stress equilibrium in the foundation soil and determine the soil lateral pressure coefficient to obtain initial stresses that closely matched the field measurements. The lateral pressure coefficient of the soil not only depends on its properties and density but also its stress history. The relationship between the lateral pressure coefficient (K) and the Poisson's ratio ($\mu$) can be approximately expressed as K = $\mu$/(1 − $\mu$). In this experiment, the values

of K for the two types of foundation soil were 0.49 and 0.45, respectively. Once the stress equilibrium in the ground was achieved, uplift and horizontal stresses were applied at the top of the pile. Convergence analysis based on the maximum unbalanced force ratio indicated that setting a maximum unbalanced force convergence ratio of $1 \times 10^{-5}$ had no significant impact on the results.

**Table 3.** The value of soil and pile parameters in numerical simulation.

| Name | Depth /m | Density /(g/cm$^3$) | Elastic Modulus /MPa | Poisson Ratio | Cohesion /kPa | Internal Friction Angle/(°) |
|---|---|---|---|---|---|---|
| Silty gravel | 0~3 | 2.09 | 4.9 | 0.32 | 13.70 | 27.03 |
| Silty sand | >3 | 2.13 | 8.2 | 0.30 | 21.39 | 16.77 |
| Pile body | \ | 2.50 | 31,500.0 | 0.20 | \ | \ |
| Pile cap | \ | 2.50 | 31,500.0 | 0.20 | \ | \ |
| Upper columns | \ | 2.50 | 31,500.0 | 0.20 | \ | \ |

**Table 4.** The value of interface parameters in numerical simulation.

| Name | Shear Stiffness | Normal Stiffness | Cohesion/kPa | Internal Friction Angle/(°) |
|---|---|---|---|---|
| Interface 1 | $7 \times 10^9$ | $7 \times 10^9$ | 10.96 | 21.62 |
| Interface 2 | $7 \times 10^9$ | $7 \times 10^9$ | 17.11 | 13.42 |

*3.2. Comparison and Analysis of Experimental and Inversion Results*

Numerical inversion was performed on the field test of the pile group foundation, and the simulation results are presented as displacement contour plots in Figure 13. A comparison between the numerical simulation and experimental results is shown in Figure 14.

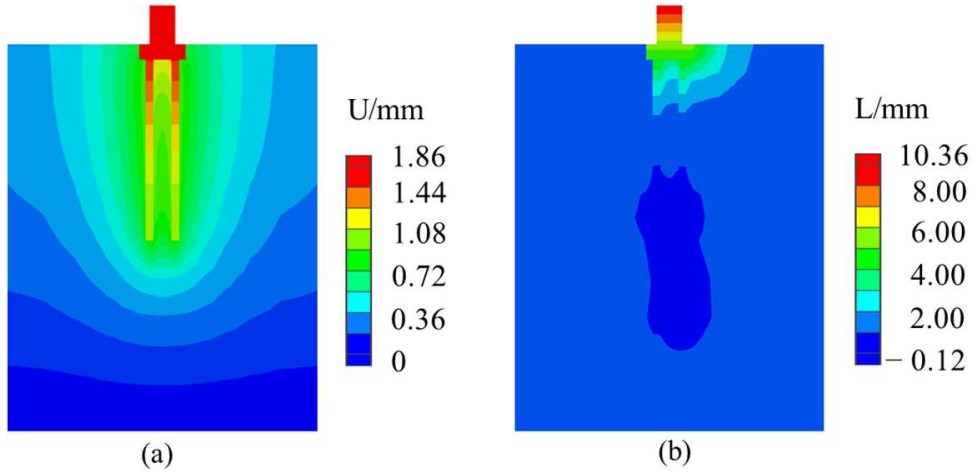

**Figure 13.** Displacement cloud of numerical inversion: (**a**) carried uplift load displacement cloud map; (**b**) carried horizontal load displacement cloud map.

From the displacement contour plot in Figure 13, it can be observed that in the horizontal resistance test, most of the soil displacement occurs in the upper layers of the soil adjacent to the pile, and that the pile cap also experiences significant horizontal thrust, leading to large displacements. The pile body exhibits a certain amount of deflection, and the lower soil experiences displacements in the opposite direction to the upper soil due to the compression from the pile. In contrast, in the uplift resistance test, only the pile cap side provides lateral frictional resistance, resulting in a significant reduction in the pile cap's contribution (consistent with the analysis of the field test results). The displacement distribution of the soil adjacent to the pile is more uniform in the uplift test.

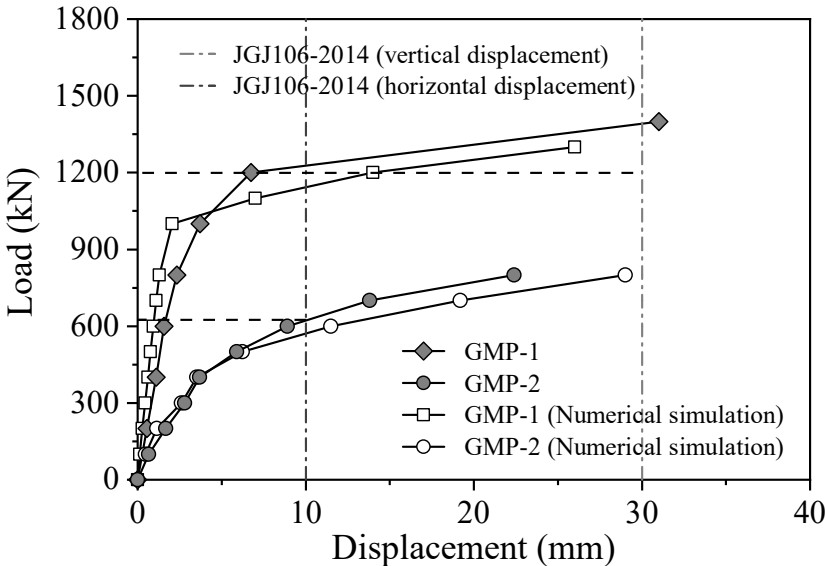

**Figure 14.** Comparison of load–displacement curves between test and numerical simulation.

From Figure 14, it is evident that the inversion results are in relatively close agreement with the experimental results. However, there are still some numerical discrepancies. The reasons for these differences can be attributed to several factors: in numerical simulations, contact elements are assigned stiffness to simulate the gaps between the pile and adjacent soil, but in reality, the "contact stiffness" between the pile and soil decreases with the application of load. Additionally, factors such as mesh accuracy, model size, and the stress–strain relationship between the pile and soil can also influence the simulation results. Furthermore, during the on-site testing process, there may have been soil disturbances and errors in the reaction piles compared to the test piles. As a result, there is some deviation between the two, but overall, the observed trends are consistent. This indicates that the numerical simulation results can reasonably reflect the actual conditions of the micropile group, providing a solid foundation for the subsequent simulations in this study.

## 4. Study on Horizontal Uplift Combined Loads

This experiment considers the influence of both horizontal and uplift loads on the equal-sectioned micropile group foundation. However, in practical engineering applications, tower foundations rarely experience isolated uplift or horizontal loads; instead, they are commonly subjected to different directional loads simultaneously. Therefore, in the design of the foundation, the coupling effect of vertical and horizontal loads should be taken into account. However, the current design methods recommended by existing codes have certain limitations [38]. By comparing them with the field test results, numerical simulations can provide a comprehensive and intuitive understanding of the load effects on the foundation's bearing capacity. Therefore, in this study, numerical simulation will be employed to investigate the performance of the equal-sectioned micropile group foundation under various combinations of loads. Figure 15 shows the magnified deformation cloud plot of the model after applying inclined loads, with a vertical-to-horizontal (U–H) load ratio of 2:1.

From Figure 15, it can be observed that applying combined loads to the equal-sectioned micropile group results in two different directional forces acting on the pile top. In addition to the uplift deformation, lateral displacement occurs, causing soil uplift deformation on one side of the foundation. The piles also exhibit significant flexural deformation. Failure occurs in the horizontal direction at a load of 420 kN, indicating that the ultimate uplift load is determined as 840 kN.

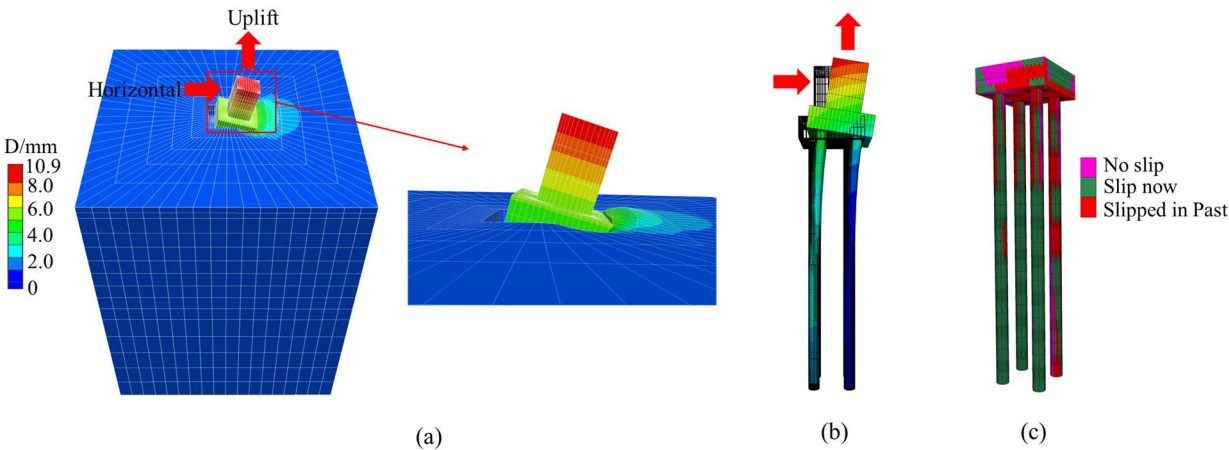

**Figure 15.** Numerical simulation effect under combined load (uplift and horizontal load ratio = 2:1): (**a**) soil deformation enlarged cloud diagram; (**b**) micropile group foundation deformation enlarged cloud diagram; (**c**) sliding cloud diagram of contact surface.

### 4.1. Pile Strain Response

The combined loading moment at any depth along the pile shaft [39] can be expressed as:

$$M = M_0 + M_1 - M_2 \tag{2}$$

where $M_0$ represents the pile shaft moment under isolated horizontal loading, $M_1$ accounts for the additional moment generated by the imbalanced lateral frictional resistance on the left and right sides of the pile section, and $M_2$ considers the extra moment resulting from the P–Δ effect. From the equation, it is evident that the influence of uplift loads on horizontal loading primarily depends on the magnitude of $M_1 - M_2$.

Figure 16 shows the distribution curves of relative strains on both sides of the micropile body under individual and combined loads (H = 1/2U) during the simulation. Figure 17 illustrates the concept of the additional moment in the micropile group. Comparing Figure 16a–c, it can be observed that after applying the horizontal load, the range of relative strain variations in the pile body is mainly concentrated in Part 1 and Part 2. The relative strain changes in the lower part of the pile body are relatively small, and the variation range can be roughly divided into two regions, labeled as Zone I and Zone II. In Zone I, for the micropile foundation subjected to the combined load, the rate of relative strain variation on the right side of the pile body is significantly higher than that of the individual uplift pile, while on the left side of the pile body, it is smaller than that of the individual uplift pile. Because the horizontal load is applied from left to right, the soil on the right side of the pile shaft experiences compression, leading to stress expansion. This results in an increase in lateral frictional resistance on the right side of the pile. Conversely, on the left side of the pile, where it is farther away from the soil, the reduced contact area between the pile and the soil leads to a decrease in lateral frictional resistance (Figure 17). Meanwhile, it can be observed that at the junction between Zone I and Zone II, the relative strain on the right side of the pile body is smaller than that of the individual uplift pile, while on the left side of the pile body, it shows the opposite trend. This is mainly due to the additional bending moment $M_1$ generated by the imbalance in lateral soil resistance on the left and right sides of the pile cross-section. The additional bending moment $M_2$ generated by the P–Δ effect is smaller than $M_1$, resulting in a net bending moment in the direction of $M_1$. Additionally, due to the pile's deflection deformation, the displacement directions in Zone II and Zone I are opposite, causing opposite patterns in the relative strain distribution along the pile body. However, comparing the different-sided piles of the group pile foundation, considering the direction of horizontal forces, the right-side pile is closer to the soil under compression on the pile body side, and the relative displacement δ of the pile is smaller. As a result, the additional bending moment $M_1$ generated by the imbalance in lateral soil

resistance is smaller for the right-side pile. Consequently, the relative strain variations on both sides of the junction between zones are smaller for the right-side pile. Conversely, the left-side pile, being farther from the soil under compression, experiences larger $M_1$, leading to greater relative strain variations on both sides of the junction.

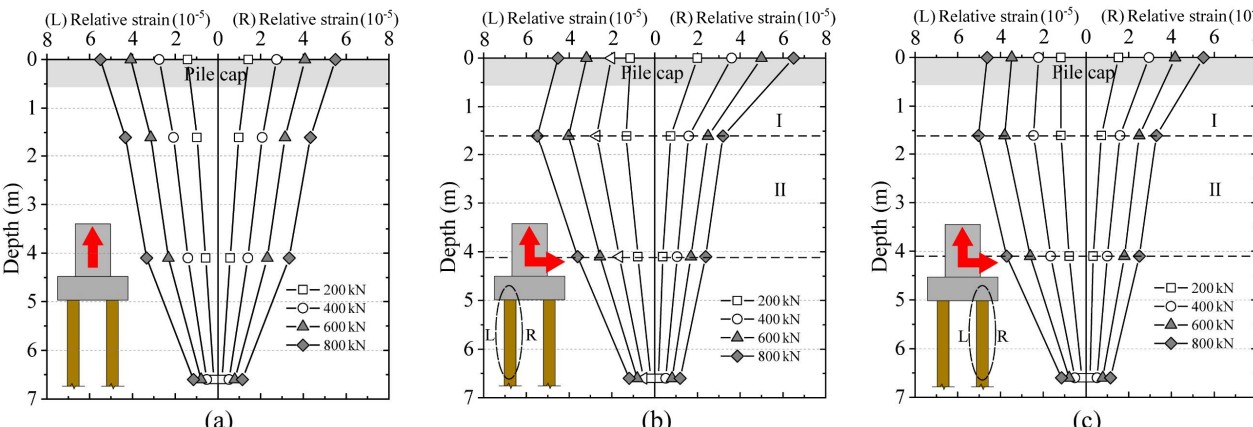

**Figure 16.** The relative strain distribution curve of the left and right side of the pile body: (**a**) H = 0; (**b**) the left foundation pile with H = 1/2 U; (**c**) the right foundation pile with H = 1/2 U. Note: The strain influence region of the pile is divided into two zones, designated as Zone I and Zone II.

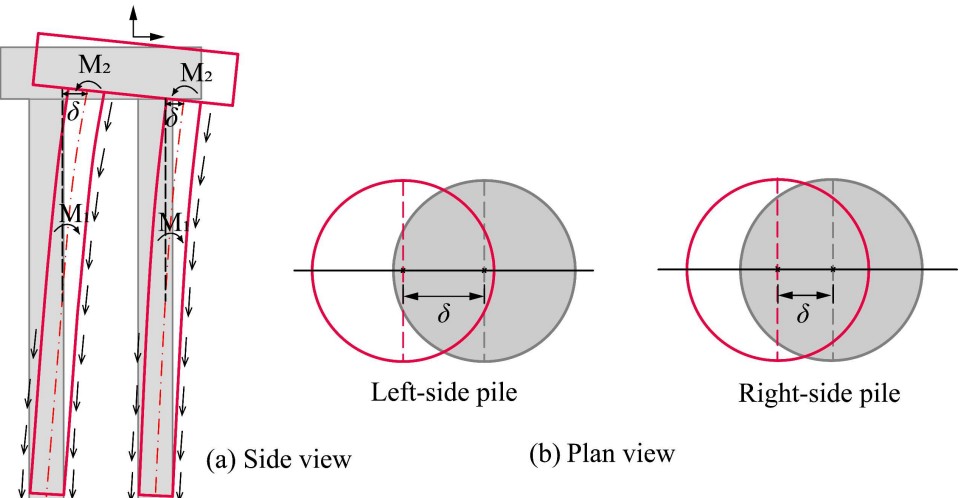

**Figure 17.** Schematic representation of additional bending moment of the pile shaft.

### 4.2. Distribution of Average Lateral Frictional Resistance

Considering that pile shaft strain to some extent reflects the distribution of axial forces within the pile shaft and that the slope of the pile shaft axial force curve represents the magnitude of lateral frictional resistance, this section focuses on analyzing variations in lateral frictional resistance along the pile shaft. The average lateral frictional resistance of the pile shaft is calculated based on the rate of relative strain variation along the pile shaft. Figure 18 illustrates the distribution curves of the average lateral frictional resistance for individual vertical loading and combined loading (H = 1/2U) applied to different foundation piles.

From Figure 18, it is evident that the average lateral frictional resistance of the ultimate pile body within the Part 1 range is 29.7 kPa, 29.5 kPa, and 22.7 kPa for the foundation piles without applying horizontal loads, the left foundation pile, and the right foundation pile after applying horizontal loads, respectively. With the increase in combined loads, the average lateral frictional resistance of the pile body within the Part 2 range significantly increases, while within the Part 3 range, the average lateral frictional resistance slightly

decreases. When compared to the right foundation pile, the left foundation pile exhibits a greater enhancement in the average lateral frictional resistance within Part 2, and a more pronounced reduction in the average lateral frictional resistance within Part 3. This behavior is attributed to the deformations and bending of the pile body.

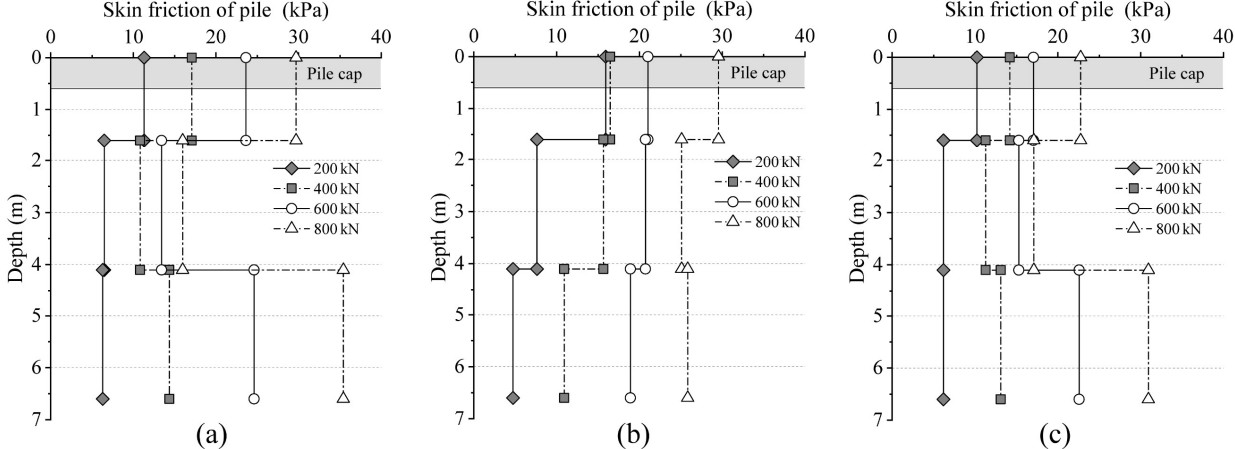

**Figure 18.** Distribution curve of average frictional resistance of foundation pile: (**a**) H = 0; (**b**) the left foundation pile with H = 1/2 U; (**c**) the right foundation pile with H = 1/2 U.

### 4.3. Bearing Characteristics under Uplift–Horizontal Load Interaction

In practical engineering applications, the proportion of uplift load to horizontal load may vary. To investigate the influence of combined loads on the bearing performance of the micropile group, further simulations were conducted by applying different ratios of uplift and horizontal loads. The loading ratios and the corresponding failure loads are presented in Table 5.

**Table 5.** Different loading ratios and failure loads of U and H.

| Ratio of Uplift to Horizontal Load | | 10:1 | 5:1 | 4:1 | 3:1 | 2:1 | 1:1 | 2:3 | 1:2 |
|---|---|---|---|---|---|---|---|---|---|
| failure load /kN | uplift | 1030 * | 1020 * | 996 * | 950 * | 840 | 560 | 395 | 304 |
| | horizontal | 105 | 204 | 249 | 316 | 420 * | 560 * | 592 * | 608 * |

Note: * loads represent the failure loads and loads without * represent the corresponding values of the other direction load at the failure load ratio.

Figure 19 shows the load–displacement curves of the pile foundation under different ratios of uplift and horizontal (U–H) combined loads. Figure 20 presents the polynomial fitting results obtained using matlab based on the discrete data obtained from numerical simulations.

As shown in Figure 19, under the influence of horizontal loads, the uplift capacity of the micropile group gradually decreases with the reduction in the ratio of uplift-to-horizontal-load application. This indicates a noticeable weakening effect of applying horizontal loads on the uplift bearing capacity of the pile group under similar soil conditions. This finding aligns with the conclusions drawn by Sharma and Buragohain [23] from indoor model experiments when subjecting micropile groups to loads at different angles. Under the influence of uplift loads, when the ratio (n) is 0.76, the horizontal bearing capacity of the micropile group is similar to that of applying horizontal load alone, but when n is greater than this value, the horizontal bearing capacity of the micropile group is weakened. On the other hand, when n is less than this value, with the decrease in the proportion of uplift load, the horizontal ultimate bearing capacity of the micropile group slowly increases and gradually stabilizes. At this point, the uplift load has a positive effect on the horizontal bearing capacity. Based on Table 4, it can be observed that when n ≥ 3, relative sliding occurs between the pile and soil, and the uplift bearing capacity reaches its limit first,

while the horizontal load is relatively small and does not meet the failure criterion. When $3 > n \geq 1$, the load–displacement curve exhibits a "steep transition" type, with a larger proportion of horizontal load, and the pile foundation undergoes horizontal failure first. When $n < 1$, both the uplift and horizontal bearing capacities of the micropile group are enhanced. However, due to the relatively small increase in horizontal bearing capacity, the uplift bearing capacity is not fully exploited. At this point, the overall bearing capacity of the micropile group is still determined by the horizontal bearing capacity.

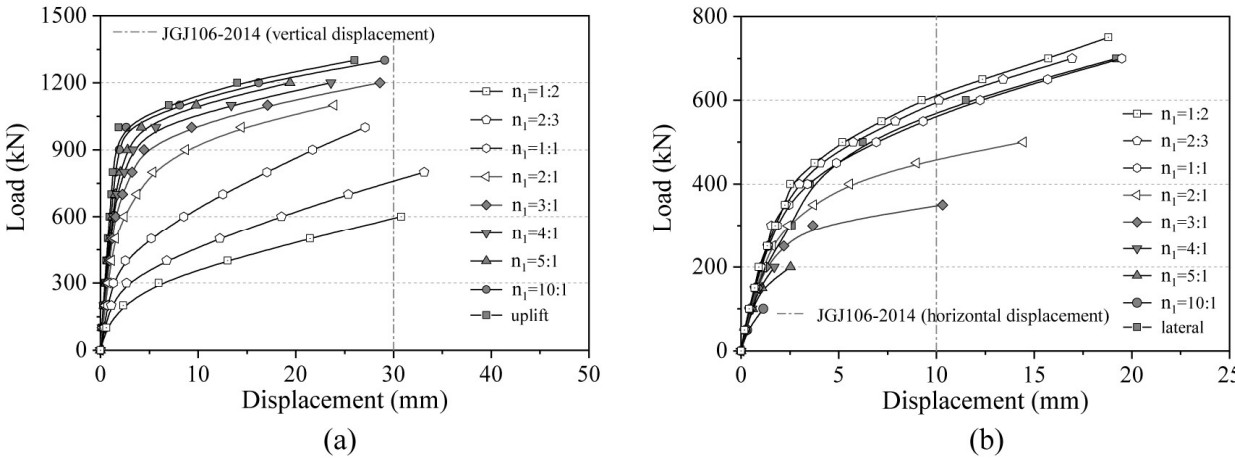

**Figure 19.** Load–displacement curves of micropile group foundation with different load ratios: (**a**) uplift load−displacement curve; (**b**) horizontal load−displacement curve.

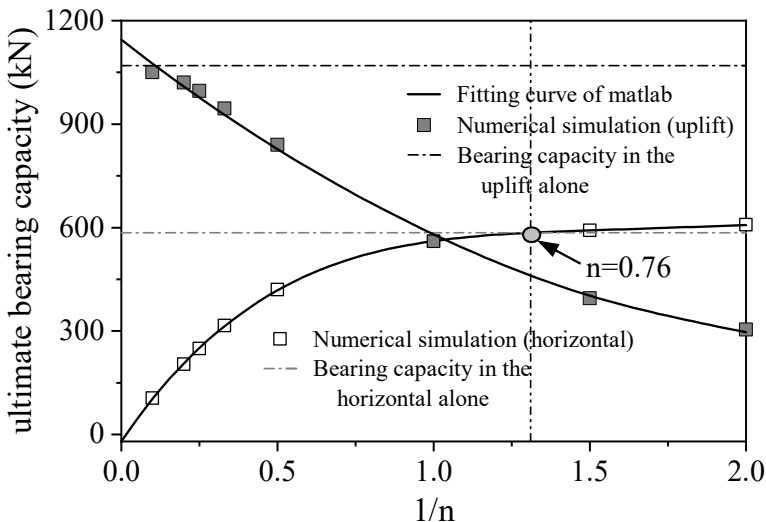

**Figure 20.** Load–load ratio reciprocal curve of micropile group foundation.

The results in Figure 20 indicate that under U–H combined loading, the ultimate uplift bearing capacity of the pile foundation approximately follows a quadratic function relationship with $1/n$, while the ultimate horizontal bearing capacity approximately follows a quartic function relationship with $1/n$. The expressions for these relationships are given by Equations (3) and (4), respectively, and the mean squared error function for the fitted curves is expressed as Equation (5).

$$U = 140.6\left(\frac{1}{n_2}\right)^2 - 705.2\left(\frac{1}{n_2}\right) + 1144.6 \tag{3}$$

$$H = -57.8 \left(\frac{1}{n_2}\right)^4 + 418.9 \left(\frac{1}{n_2}\right)^3 - 1119.2 \left(\frac{1}{n_2}\right)^2 + 1339.1 \left(\frac{1}{n_2}\right) - 20.0 \qquad (4)$$

$$E = \frac{1}{n} \sum_{i=1}^{n} (y_i - Y_i)^2 \qquad (5)$$

where $E$ represents the mean squared error of the fitted curve, $y_i$ denotes the fitted values, $Y$ represents the actual values, $E_U = 265.9$, and $E_H = 2.4$. The fitting results show good agreement between the fitted values and the actual values.

Figure 21 shows the load-yield envelope of the micro pile group foundation under the U–H load plane, where all the yield points are connected. Regression analysis of the experimental data reveals that the horizontal and vertical bearing capacities under combined loading exhibit an approximate 1/4 elliptical relationship. A comparison with previous studies [40–42] indicates that the U–H bearing capacity envelope obtained in this study is relatively close to the Koumoto empirical formula results. However, there are still certain deviations between them when the horizontal load is relatively small. Therefore, for the pile foundation under combined loading, the relationship between the horizontal and vertical loads is not exactly the same. Factors such as pile structure, geological conditions, and loading sequence should be thoroughly considered to establish corresponding criteria for assessment.

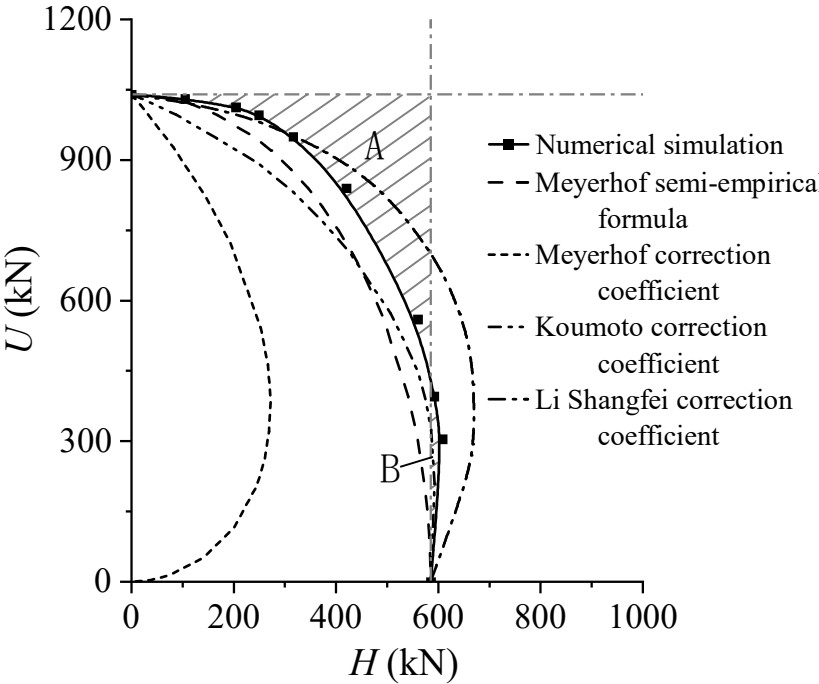

**Figure 21.** Spatial yield envelope of micropile group foundation.

In Figure 21, it is evident that the spatial yield envelope of the micropile group foundation under U–H loading, along with the vertical line representing the ultimate horizontal and uplift bearing capacity under separate loading conditions, encloses two distinct regions, denoted as A and B. The loading conditions within region A may lead to damage of the pile foundation, while those within region B are relatively safe. This indicates that the bearing characteristics of the micropile group foundation differ under combined loading compared to separate loading. Therefore, when designing micropile group foundations, careful consideration should be given to the influences of these two distinct regions.

## 5. Conclusions

(1)  During the field uplift loading test of micropile group foundations in the plateau mountainous area, the load distribution among individual piles is relatively uniform. The load is transmitted from top to bottom until it reaches the ultimate limit state. The lateral friction resistance of the pile body in the micro pile group foundation follows the order: Part 3 > Part 1 > Part 2.

(2)  The micropile group foundation has an ultimate uplift bearing capacity of 1200 kN with a steep load–uplift displacement curve. Its horizontal bearing capacity is 620 kN with a gradual load–horizontal displacement curve. The numerical inversion results are in good agreement with the experimental data, confirming the feasibility of numerically studying the bearing behavior of micro pile group foundations.

(3)  Under the combined load–coupling effect, the additional moment $M_1$ generated by the unbalanced lateral friction resistance on both sides of the pile cross-section is greater than the additional moment $M_2$ generated by the P–$\Delta$ effect. As a result, the relative displacements and lateral friction resistance on the left and right sides of the upper part of the foundation are different from those under individual uplift loading conditions, and the piles located farther from the soil under compression exhibit larger net bending moments.

(4)  Compared to the condition of applying uplift load alone, the flexural deformation of the pile caused by the application of combined loads results in an increase in lateral friction resistance on the Part 2 side, while the lateral friction resistance on both the Part 1 and Part 3 sides decrease. As a result, the overall lateral friction resistance of the pile decreases, leading to a reduction in the ultimate uplift bearing capacity of the foundation. Moreover, the larger the horizontal load, the more significant the reduction effect.

(5)  The bearing performance of the micropile group foundation under U–H combined loading is related to the load ratio parameter, n. As n decreases, the uplift bearing capacity of the micropile group foundation continues to weaken, while the horizontal ultimate bearing capacity gradually increases. When n = 0.76, the horizontal bearing capacity of the foundation becomes equivalent to the horizontal bearing capacity under the action of a single horizontal load.

(6)  Based on regression analysis of experimental data, the horizontal bearing capacity follows a fourth-order function relationship with the reciprocal of the load ratio, while the uplift bearing capacity exhibits a second-order function relationship with the reciprocal of the load ratio. These bearing characteristics differ from those observed under single-direction loading. The enclosed area between the yield envelope and the plumb line of unidirectional ultimate bearing capacity under the combined load is divided into the "failure zone" and "safety zone." The reduction in uplift capacity within the "failure zone" should be considered during the pile foundation design.

Research has shown that micropile group foundations, when used as tower foundations in plateau mountain areas, exhibit favorable load-bearing performance. However, the study of the foundation's load-bearing capacity under combined loading conditions is essential and should not be overlooked. In most cases, combined loading tends to weaken the load-bearing performance of micropile groups. This should be given due consideration in engineering practice. While the results of this study provide valuable insights into the effects of combined loading, it is important to note that they should be further validated due to limitations in constitutive relationships and boundary conditions within the numerical simulations. Additionally, there is a need for more in-depth research on the influencing factors and sensitivity analysis of combined loading on micropile groups, which currently remains a topic for future investigation.

**Author Contributions:** All authors contributed to the study conception and design. Material preparation and data collection were performed by H.L. and G.R. Data analysis and the first draft of the manuscript was written by H.L., and all authors commented on subsequent versions of the manuscript. All authors have read and agreed to the published version of the manuscript.

**Funding:** This research was funded by the State Grid Sichuan Electric Power Company (SGSCDZ00JSJS2100272).

**Institutional Review Board Statement:** Not applicable.

**Informed Consent Statement:** Not applicable.

**Data Availability Statement:** Not applicable.

**Acknowledgments:** The valuable comments made by the anonymous reviewers are sincerely appreciated.

**Conflicts of Interest:** The authors declare no conflict of interest.

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
