# Peer review of "Horizontal and Uplift Bearing Characteristics of a Cast-In-Place Micropile Group Foundation in a Plateau Mountainous Area"

_sustainability, doi:10.3390/su151813554_

Round 1
Reviewer 1 Report
Title: "Horizontal and Uplift Bearing Characteristics of Cast-in-Place Micropile Group Foundation in Plateau Mountainous Area"
The authors have conducted intriguing experimental tests to investigate the horizontal and uplift bearing characteristics of cast-in-place micropiles.
Comments and Suggestions:
1. Could the results of the horizontal and uplift bearing capacities be compared with traditional methods such as conventional equations? Adding a comparative analysis with established methods could enhance the understanding of the uniqueness and applicability of the findings.
2. Regarding Table-1, could you clarify the source of reference for the data presented? Were basic geotechnical tests conducted as part of this research, or were the data derived from prior studies? Providing a clear reference for the table's data origins will enhance the transparency and credibility of the presented information.
3. In terms of FEM modeling, could you consider providing a comprehensive table detailing the specific properties of each material used in the modeling? This addition would provide a more robust insight into the model's construction and allow readers to better comprehend the simulation process.
4. Given the close alignment between the trends observed in FEM modeling and experimental results, can the FEM model be relied upon for future simulations? Expanding on the reliability and potential limitations of the FEM model would provide readers with a better understanding of its predictive capabilities.
5. Could you elaborate on the underlying concept or reference utilized for determining the failure moments of the piles? This information would clarify the basis for the failure analysis and enhance the paper's technical depth.
6. It might be insightful to compare the study's findings with existing 1-g tests of small-scale piles conducted in laboratory settings. Numerous papers have simulated similar processes in laboratory conditions. Such a comparison could contextualize the research findings and strengthen their real-world implications.
7. Is there a geotechnical guideline or rule available for estimating the required displacements to induce pile failures? Exploring whether existing guidelines apply to the research's context could offer valuable insights into the practical implications of the study.
Author Response
Dear Reviewer,
We would like to express our sincere gratitude for taking the time to review our manuscript. Your valuable comments and suggestions have been incredibly insightful and have significantly contributed to improving the quality of our paper.
We have carefully addressed each of your points and made the necessary revisions to address the concerns you raised.
"Please see the attachment."

Reviewer 2 Report
Dear Authors,
. Suggested corrections;
I think that abstract is a little too much. It should include the importance of the study, briefly the procedures performed and the important results obtained. Therefore, I suggest to rewritten abstract.
In this context, it would be useful to add a paragraph about soil-foundation-structure interaction. Because local soil characteristics will change this interaction especially under seismic loads. Some suggested references and similar references could be used. Suggested references;
Assimaki, D., et al (2005). Effects of local soil conditions on the topographic aggravation of seismic motion: parametric investigation and recorded field evidence from the 1999 Athens earthquake. Bulletin of the Seismological Society of America, 95(3), 1059-1089.
Avcil, et al., (2022). The effect of local soil conditions on structure target displacements in different seismic zones. GümüÅŸhane Üniversitesi Fen Bilimleri Dergisi, 12(4), 1000-1011.
Çelebi, E., et al. (2012). Non-linear finite element analysis for prediction of seismic response of buildings considering soil-structure interaction. Natural hazards and earth system sciences, 12(11), 3495-3505.
In addition, I think it would be useful to add a few resources about the behavior of structures on sloping ground.
Işık et al., (2020). Eğimli Zeminlerde İnşa Edilen Betonarme Binaların Deprem Davranışlarının İncelenmesi. Avrupa Bilim ve Teknoloji Dergisi, (20), 162-170; Mohammad, et al. (2017). Seismic response of RC framed buildings resting on hill slopes. Procedia engineering, 173, 1792-1799.
At the end of the introduction, please write the steps in your article and show the difference / novelty of the article more clearly, especially from similar compilation studies.
Please, improve the quality of the Figures.
If it can be possible, please add some pictures from experimental studies.
Please expand literature part a little more using some studies from other countries.
It would be very good if the authors can compare the similar studies with their comments, if possible. Comparative comments will be much more interesting to readers.
Please include the contribution of your work to similar studies in the future.
Include the limitations of your work in the conclusion.
Yours Sincerely
Author Response

(The authors gave the same response as above.)

Reviewer 3 Report
The following suggestions are helps to improve the quality of the manuscript.
1. In Introduction section add latest literature studies with respect to cast-in-place micropile group foundation.
2. In table-1, Why same depth is chosen for sample id 1 & 2.
3. In Figure-6 GMP-1 horizontal displacement after 1200kN displacement is huge. With respect to this technical explanation must be added in this section.
4. In micropile group load carrying capacity part-3 shows huge variation. technical explanation must be added which factors responsible to cause this huge variation.
5. In numerical modelling it is advised to add separate paragraph with respect to type of modelling, mesh ratio adopted in modelling, boundary conditions and other details of modelling details adopted.
6. Stress analysis is also one of the major parameters helps to study the effect of load mechanism. It is suggested to add in separate section.
Moderate editing of English language required
Author Response

(The authors gave the same response as above.)
